# Healthcare Wearable Sensors Adhesion to Human Fingernails and Toenails

**DOI:** 10.3390/mi15010069

**Published:** 2023-12-29

**Authors:** Katsuyuki Sakuma, Leanna Pancoast, Yiping Yao, John Knickerbocker

**Affiliations:** 1IBM Thomas J. Watson Research Center, Yorktown Heights, NY 10598, USA; leanna.pancoast@outlook.com (L.P.); knickerj@us.ibm.com (J.K.); 2IBM Corporation, Infrastructure, Hopewell Junction, NY 12533, USA; yyao@us.ibm.com

**Keywords:** healthcare, wearable sensors, nail glue, human fingernails, toenails, Parkinson’s disease

## Abstract

A novel adhesion method of a sensor to a fingernail is described. Wearable sensors can provide health insights to humans for a wide variety of benefits, such as continuous wellness monitoring and disease monitoring throughout a patient’s daily life. While there are many locations to place these wearable sensors on the body, we will focus on the fingertip, one significant way that people interact with the world. Like artificial fingernails used for aesthetics, wearable healthcare sensors can be attached to the fingernail for short or long time periods with minimal irritation and disruption to daily life. In this study the structure and methods of healthcare sensors’ attachment and removal have been explored to support (1) the sensor functional requirements, (2) biological and environmentally compatible solutions and (3) ease of attachment and removal for short- and long-term user applications. Initial fingernail sensors were attached using a thin adhesive layer of commonly available cosmetic nail glue. While this approach allowed for easy application and strong adhesion to the nail, the removal could expose the fingernail and finger to a commercially available cosmetic nail removal (acetone-based chemical) for extended times measured in minutes. Therefore, a novel structure and method were developed for rapid healthcare sensor attachment and removal in seconds, which supported both the sensor functional objectives and the biologically and environmentally safe use objectives.

## 1. Introduction

As computing technology, battery technology, and sensor technologies improve, new miniaturized healthcare micro-systems can be applied to different places on or inside the body. Advancements in continuous monitoring of human health (i) for wellness and (ii) disease onset and progression, and (iii) from injury, surgery and rehabilitation provide patients with opportunities to improve patient care and quality-of-life (QOL). Research studies and future patient adoption of wearable and implantable sensors for locations on or in the body may provide new health insights while being less obtrusive for the patient [1,2]. Many micro-systems and health sensors will be temporarily attached to the body to allow for replacement or to prevent harmful physical responses. These wearable health sensors require a patient friendly means of temporary use and removal. Some temporary attachment methods use structural or mechanical means to attach the device to the body, such as the band of a smartwatch with health sensors that secures the device to the wrist or hearing aids with health sensors that are worn in the ear. Some methods require the use of an adhesive to remain in place on the body, such as a wearable insulin pump [3]. These methods allow for non-harmful attachment and removal of the wearable healthcare sensors on locations of the body that minimize interference of the patient’s normal movements. Different places on the body will require different methods of attachment to allow for ease of use. Our study focused on the attachment and removal of electronic health sensors to the human fingernail and toenail.

The fingernail deforms in many tasks that are done every day, from fine finger touch to more forceful and directional finger movements, such as gripping, writing with a pen or pencil, opening a door, picking up a cup of coffee, brushing your teeth, pressing the button in an elevator or other activities in daily living. Precision monitoring of nail deformations allows insight into how the wearer interacts with everyday objects and performs daily activities, which can then provide useful health insights into different diagnoses [4]. For example, the recovery of hand and finger movement after a surgery can be tracked with precise, consistent force and movement data. The effectiveness of medication that might affect grip strength can also be tracked. Monitoring everyday tasks where fingertip and hand movements send precise force and movement data streams requires the sensor to be attached securely to the fingernail during use and to be easily removed thereafter. This paper will focus on a new method of adhesion of a strain gauge to a human nail to measure nail deformation, as previously reported by Sakuma et al. [5,6,7]. The demonstrated new structure, and method are shown below in Figure 1 and Figure 2 respectively. The new approach provided ease of attachment and removal and supported good functional data from the sensor. While this new method was demonstrated with the strain gauge, other health sensors, microsystems, and components such as antennae, micro-controllers, and micro-batteries could be attached.

## 2. Related Work

Prior publications using electronic and optical sensors examining fingertip movement have been reported. Mascaro et al. developed a fingernail sensor which can detect changes in fingernail color patterns depending on blood distribution under the nail using an optical reflectance photo plethysmograph sensor [8,9]. They attached the sensor to the fingernail using a double-sided transparent adhesive. This method is possible with sensors such as LEDs, but is not suitable for strain sensors that need to detect the actual deformation of the nail on contact, which changes only in micrometers. Kandori et al. developed a measurement tool that quantitatively detects finger movement from a magnetic sensing system by attaching a magnetic induction coil and a sensing coil to the thumb and index finger with double-sided tape [10]. They measured the velocity and acceleration of the finger movement during finger tapping, and they used it as a quantitative method. The method of attaching a sensor to a finger using tape is suitable only for sensing the movement of relatively large objects, such as a finger. Serina et al. studied the force response of fingertip pulp to compression under quasi-static conditions by using geometry, material inhomogeneity, skin nonlinearity, and skin stretch conditions [11]. Although the LEDs were attached to the surface of the nail, they did not describe the specific method or conditions under which the sensor was attached to or detached from the nail. Sakai et al. investigated the distribution of nail strain of the right index finger while the fingertip was compressed with a force less than 14 N [12]. Cyanoacrylate adhesive was used to attach the sensor to the nail, but there is no study on the removal method [13].

Various sensors attached to fingertips to detect fingertip movement have been studied. However, none of the above studies examined the details of sensor attachment and removal methods, process flow, or materials. In addition, with respect to previous studies measuring nail deformations, no research has been conducted to date on the quick attachment and removal of sensors. Since monitoring daily life based on nail deformations using a wearable sensor requires a wearable sensor that is easy to attach and detach, this study evaluated the adhesion of wearable sensors to human nails and proposed a new method of attaching sensors to nails.

## 3. Sensor Attachment Structure and Methods

### 3.1. Baseline Method to Attach Strain Gauge to Nail

The baseline attachment method of the strain gauge used nail glue, which was based on the material cyanoacrylate [14]. This is more commonly known as super glue. It is a fast-drying, strong-bond material. While this allows for a very fast (<1 min) application of the sensor to the nail, the removal of the sensor required either a strong mechanical separation causing harm to the wearer and sensor or could expose part of the wearer’s finger to the risk of prolonged exposure to chemical solvents, potentially damaging the finger and nail over time [15]. A goal to reduce the risk of exposure to chemical solvents from about 4 to 20 min to shorter times hopefully measured in seconds and with more biologically and environmentally friendly solvents was targeted for sensor attachment and removal.

### 3.2. New Method to Attach Strain Gauge to Nail

The new sensor attachment approach used an alternate adhesive layer structure and method, as shown in Figure 2. The structure was composed of two adhesives, a slow cure, easy to remove release layer resting on the nail, and a fast-curing adhesive layer attaching the electronics to the release layer. The release and adhesion layers were applied to the nail, followed by the sensor. This paper reports on the results of the release and adhesive layers studied for strain gauge attachment to a fingernail or toenail. Several key challenges to support the target application were identified. First, the release and adhesive layers needed to dry within seconds to perhaps a few minutes for practical use. Second, the layers needed to adhere to the nail and sensor strongly enough to keep the sensor on the nail through a variety of different activities depending on target use from <2 h to days or longer while providing the sensors with a continuous quantitative data stream for the use application. Finally, the layers needed to be removable within seconds to a few minutes safely and easily by the patient or care provider with minimal mechanical or solvent exposure to the nail or the sensor.

A number of different structures and methods were explored, from mechanical peel removal to solvent removal on human and artificial nails. After exploration of specifications and experimentation with human and on artificial fingernails, results of the study lead to a prioritized release layer and adhesive layer for expanded testing.

## 4. Experimental Setup

We tested three different acceptable and commercially available adhesive materials on three different types of nails to determine the cure and removal times. We focused on a first air-cure, water-solvent adhesive, referred to as material A, a second UV-cure, water-solvent adhesive, referred to as material B, and a third fast air-cure adhesive known as nail glue that had been used in prior baseline assessments. Materials A and B were chosen for their ability to be mechanically peeled off the human nail. The nails used are a human nail, referred to as nail H, and two commercially available acrylic artificial nails, namely, a white plastic nail referred to as nail W and a pink plastic nail referred to as nail P. We tested the material adhesion to the artificial nails to determine the feasibility of using the artificial nails as a stand-in for human nails in longer-term or harsh environmental tests of electronics attached to the artificial nails.

Next, we applied a thin coat of each of the adhesive materials to each of the artificial nails to determine full cure time and to assess removal times. Full cure was determined by contacting the surface of the drying adhesive to inspect for any adhesive deformation. Once the cure time was determined, a new artificial nail was prepared using the same method and cure time and not disturbed until the release layer had cured.

Different removal methods were tested via a similar methodology, without attaching an adhesive to the release layer. Tweezers were used to peel the layer of adhesive away from the artificial nail. This required a controlled methodology to begin the peeling process. Once peeling started, the rest of the layer was removed easily.

Once the attachment and removal timing of the adhesives alone on the different nails were determined experimentally with repeated measurements, assessments to test the ability and timing to attach and remove strain sensors to each of the artificial nails were undertaken. These sensors’ attachments and removals, using the new two-layer structure with release and the adhesive layers and method, used biological and environmentally friendly materials as a release layer and a cyanoacrylate adhesive as a bonding layer.

Examples of experimental results are shown in Figure 3, Figure 4 and Figure 5. Figure 3 shows pictures of removing a strain sensor from the human nail, comparing two different adhesives. Figure 4 shows examples of the strain sensor removal from an artificial fingernail where the new adhesive structure permitted strain sensor removal without damage, permitting sensor reuse. Typical adhesive attachment and removal times are reported in Table 1, where the attachment and removal times can be achieved in less than a minute, respectively.

To further examine the adhesive differences between the different nails, scanning electron microscopy (SEM), Fourier-transform infrared spectroscopy (FTIR) by Bruker UMOS FTIR Spectrometer using the ATR technique, and atomic force microscopy (AFM) were performed on each of the nails. Surface roughness was measured by AFM within a 20 μm × 20 μm area on each nail. The average of the three measurements is presented in Figure 5 in the top right of the SEM images of the respective nails.

## 5. Adhesion to Nails Characteristics 

The cure and removal times of the different adhesives on the different nails are summarized in Table 1. Results show the strain gauge sensor could easily be removed from nail H and nail W with the new release layer and adhesive layer structure that had been attached to the artificial fingernail and strain gauge sensors. However, the adhesive materials chosen acted differently on different nail types. Note that material A cures slower on nails P and W than on nail H, the nail glue would not cure to nail P, and material B could not be removed from nails W or P.

To further explore the reasons why the adhesive materials act differently on the different nails, a more detailed chemical and physical understanding of the materials that make up the artificial nails and the adhesives was needed. As the supplier composition of the artificial nails had limited information, FTIR was performed on each of the artificial nails. The spectra of each nail can be seen in Figure 6. Nail H was found to have an amide I band of a protein structure, nail W was a copolymer, and analysis of nail P suggests it possibly is a poly (ester urethane). SEM and AFM analysis were also completed on each of the nails to determine the surface roughness, as shown in Figure 5. The average root-mean-square measurements (RMS) for nails H, W, and P were 244.12 nm, 36.88 nm, and 14.41 nm, respectively.

The results show that material A cured faster on human nails than on fake nails. This may be due to chemical composition differences between artificial and human nails or other factors, such as porosity in the surfaces or reaction between the surfaces and applied surface coating. The AFM results show that nail H had a higher surface roughness than both nails W and P. The higher surface roughness may provide a larger surface area and mechanical adhesion interlock for material A. The FTIR results show that nail H contains protein, and protein may provide more energy for water molecules to escape the nail surface and may have made the cure process of material A shorter.

The nail glue was not able to cure properly on artificial Nail P. This may be related to the chemical composition of nail P or the surface of this nail having a low 14.41 nm RMS surface roughness or other factors. An observation made showed that a surface with low roughness and low porosity led to poor bonding and low adhesion for the nail glue compositions tested. Material B was unable to be removed from the artificial nails W and P that were tested. This result may be due to the way material B bonds to the artificial surface chemistries, the type of bond formed, and/or the cure process. As opposed to an air cure, material B used UV light to undergo polymerization as part of its cure process. From FTIR analysis, the commercially available acrylic nails both contain methacrylate (nail W is acrylic, and nail P has an ester functional group), which each have a chemical component similar to the film former in material B. The film former in material B undergoes polymerization/cross-linking under UV light, the polymerization/cross-linking is initiated by photoinitiator in material B. Therefore, the mechanism of material B reacting with and curing with the artificial nails likely leads to material B being cross-linked with the artificial nails’ surface and perhaps to some depth on the nails W and P. Whereas for material B applied to nail H, the bonding between the two seems to create a mechanical bond but does not appear to create a chemical bond with cross-linking to the surface. In the removal process of material B from each of the artificial nails (W and P) and from nail H, results show that film removal from the human nail was much easier than from the artificial nails. This may confirm the difference in bonding mechanisms between human nails and artificial nails. We hypothesize that low or no cross-linking occurred between material B and the human nail and that a high level of cross-linking between material B and the artificial nails led to the high adhesion to the artificial nails.

## 6. Potential Application of Attaching Strain Gauge Sensor to Toenail

Patients with degenerative neurological diseases, such as Parkinson’s Disease (PD), suffer from a diverse set of symptoms. The hallmark symptoms of Parkinson’s disease are tremor, bradykinesia, rigidity, and postural instability. Patients suffer from different symptoms with different severities in a different order depending on the individual course of their disease state. To evaluate the effectiveness of a medication/therapy, the challenge is to categorize a patient’s current status as (i) ‘ON’, meaning relatively able to function given their current disease state and therapy, (ii) ‘OFF’, meaning not functioning well due to movement disorders or actually being somewhat paralyzed, and (iii) ‘ON’ with complications’, meaning able to move but with extra movement issues (typically dyskinesias or dystonia). Patients with more ‘ON’ time when given a new medication/therapy relative compared to the previous medication/therapy are considered to have benefitted from the new medication/therapy. An approach to this challenge may be to collect a continuous data stream that correlates well with the ON/OFF state in PD patients [5]. The approach must accommodate some patients’ limited ability to remember and perform explicit tasks, operate mobile phones, or wear sensors on their skin. In this study, we have used strain gauge sensors on human fingernails and/or toenails, which along with or without motion sensors attached enable health monitoring with continuous data streams sent wirelessly to mobile smartwatches, phones, and/or cloud computers. Recent strain gauge on toenail results during walking are reported below.

## 7. Deformation of Toenail

To measure the deformation of the toenail, a sensor unit was attached to the toenail of a subject, as shown in Figure 7. The sensor unit was composed of strain gauge sensors, motion sensors, an amplifier, an analog-to-digital converter (ADC), a micro-controller for wireless data transmission, and a battery. Dual-axis strain gauges connect to the circuit by a lead wire. A Wheatstone bridge circuit is utilized as a means to detect a small change in the resistance that is produced by strain gauges. The data were recorded at the sampling frequency of 100 Hz for characterization and post-processing. A block diagram showcasing the connections is shown in Figure 8.

Toenail deformation was monitored when a subject walked on a treadmill. Figure 9 shows typical results of the measured signals of the sensor during walking. There are signals from two strain gauges (transverse axis and longitudinal axis) for 180 s. About the longitudinal axis strain, positive strain means compression, and negative strain means stretching. Figure 9b,c shows the expanded signals from 125 s to 130 s and 145 s to 150 s. The broken vertical lines (green and black) denote the timing of toe off and heel strike. The strong peak was generated at the instant when the subject had toe off motion. When a foot made a landing on the ground from the heel, the tissue under the nail was pulled away from the distal end of the toe, and negative force was applied in the longitudinal and transverse directions. When the opposite foot stepped forward, the tissue under the nail was pushed to the distal end of the toe. This movement applied positive force in the longitudinal and transverse directions. Figure 9b,c shows the amplitude and frequency of the output of the strain gauge change by changing walking speed. The subject had about 40 strides per foot per minute. The peak-to-peak value of the strain amplitude decreased to 66% when a small step was taken during walking in this experiment. These findings may suggest that the sensor may distinguish the difference between a healthy person and a Parkinson’s disease patient, as well as the PD patient “On” and “Off” state.

## 8. Conclusions and Future Work

The new method of attaching a strain gauge sensor to a human nail by using a release layer material in addition to nail glue was developed to enhance user attachment and removal of the strain gauge health sensor. This advancement better accommodates the user with ease of attachment/removal as well as providing a safer attachment and removal method by removing the need for a helper chemical solvent and reducing time from minutes to seconds for the sensor. The two release layer materials’ structures demonstrated good release from a human nail but had differing results when tested on each of two different artificial nail materials. Future studies on different artificial nail materials could support alternate structures and materials for both human nails and artificial nails and could support extended use studies for electronic sensors. The test of adhesion and removal of the materials on different nails were completed within one day. Future work can evaluate attachment and removal for extended times of days, weeks, and perhaps months. Further evaluations are suggested to evaluate the adhesive quality, duration, and removal for extended times both on artificial and human nails. Care should be taken to ensure materials are chosen to reduce the risk of adverse reactions.

Attaching a strain gauge sensor to a human toenail produces strong, distinct signals with peaks for different times of a human stride. These strong signals provide useful gait analysis focused on how the foot interacts with the floor and provide new insights for patients such as those with Parkinson’s disease and patients undergoing rehabilitation after surgery or injury. Future studies with strain gauge sensors and motion sensors attached to the feet and hands will provide new and unique insights for health monitoring and quality of life enhancement.

There are many potential applications requiring the adhesion of electronics to a human nail. This new method enables future research using devices on the human nail while minimizing risk to both the patient and device.

## Figures and Tables

**Figure 1 micromachines-15-00069-f001:**
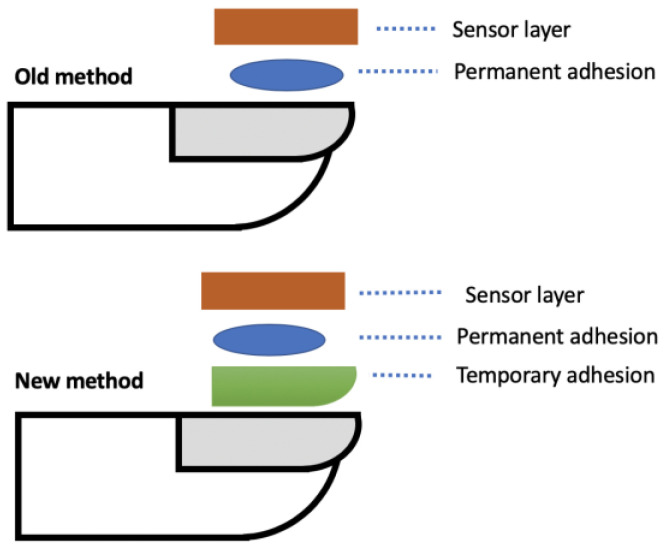
Example cross section of old and new structure and method to attach sensor to the fingernail.

**Figure 2 micromachines-15-00069-f002:**
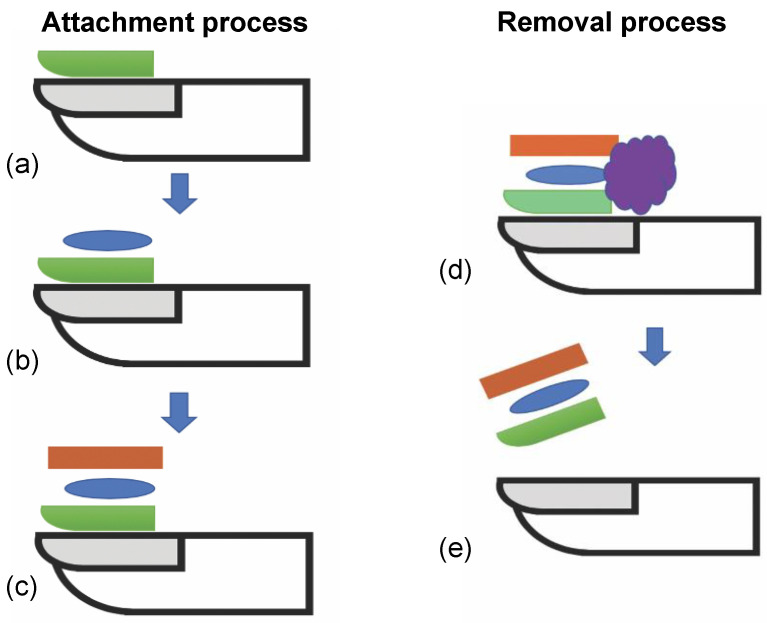
Diagram showing steps using new method for attachment and removal of sensor to nail. (**a**) Apply and cure temporary adhesive layer. (**b**) Apply permanent adhesive layer. (**c**) Attach sensor to adhesive layer and maintain pressure until cured. (**d**) Optional exposure to solvent to weaken adhesive strength. (**e**) Peel off sensor and adhesive stack from nail.

**Figure 3 micromachines-15-00069-f003:**
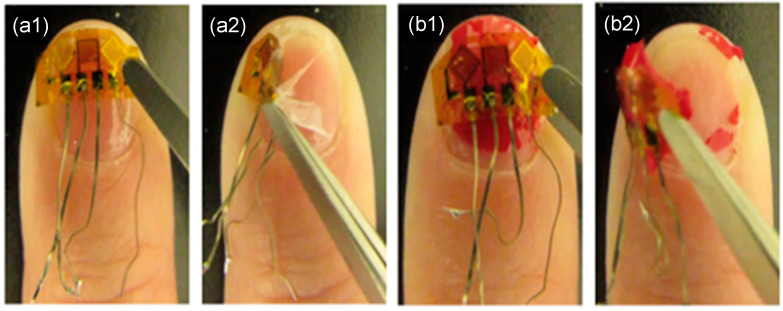
Removing strain sensor from human nail using two different materials. (**a1**) Before removal with material, (**a2**) in middle of removal of strain sensor and material A, (**b1**) before removal of strain sensor with material B, and (**b2**) in middle of removal of strain sensor with material B.

**Figure 4 micromachines-15-00069-f004:**
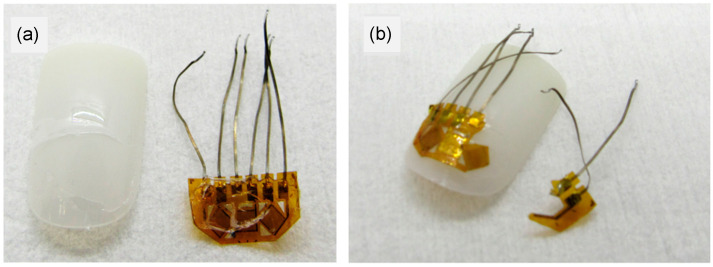
Different states of removal of strain sensor attached to artificial nails with both old and new methods. (**a**) After removal of strain sensor from artificial nail W while using new method. No damage to nail or sensor, but small amounts of adhesive A remain on both. (**b**) Old method of attaching the strain sensor to nail W. Removal results in mechanical damage to strain sensor.

**Figure 5 micromachines-15-00069-f005:**
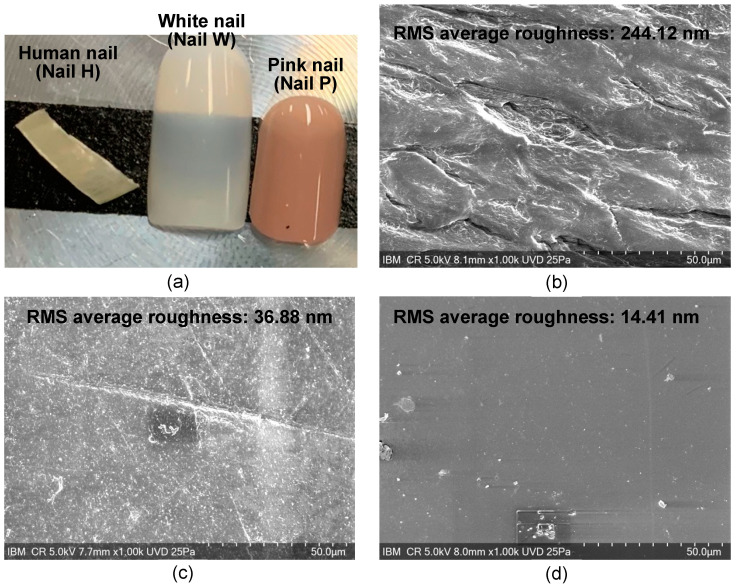
Images of different nails tested. The RMS average of the roughness of each of the nails as measured by AFM is displayed in the top of the respective SEM image. (**a**) Optical camera photo of different nails, (**b**) SEM image of human nail (Nail H), (**c**) SEM image of white nail (Nail W), and (**d**) SEM image of pink nail (Nail P).

**Figure 6 micromachines-15-00069-f006:**
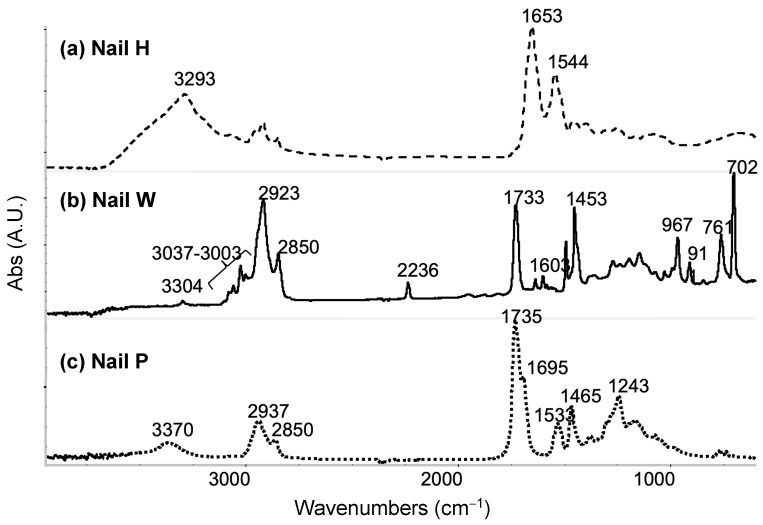
FTIR spectra for nails tested. (**a**) Nail H is found to have an amide I band of a protein structure with absorption bands at 3293 cm^−1^, 1653 cm^−1^, and 1544 cm^−1^. (**b**) Nail W is found to be a copolymer, in which it has acrylonitrile, butadiene, polystyrene, and polymethacrylate. (**c**) Nail P is possibly a poly (ester urethane) due to bands representing ester and urethane functional groups.

**Figure 7 micromachines-15-00069-f007:**
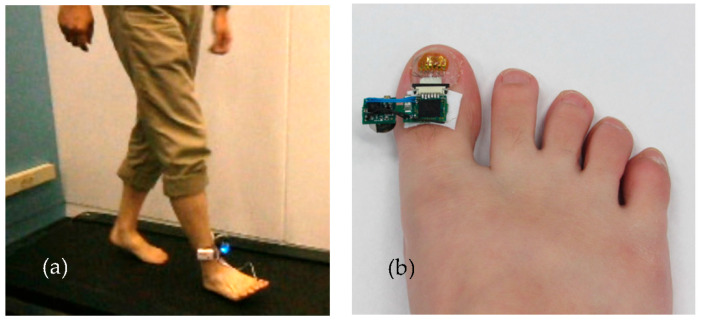
Measuring thumbnail deformity during bipedal walking by attaching a sensor to the right thumbnail. (**a**) Subject wearing strain sensor with data collection device attached to the ankle. The subject walks on a treadmill. (**b**) Wearable strain gauge sensor attached to the toenail using a new adhesive method.

**Figure 8 micromachines-15-00069-f008:**
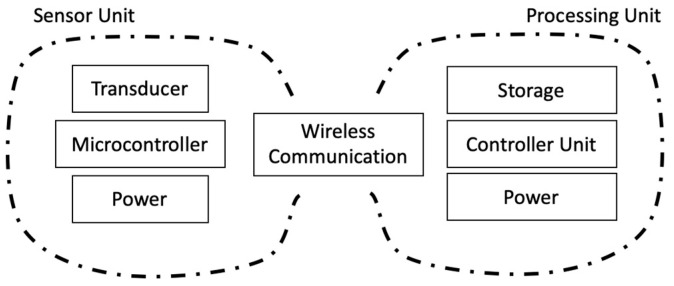
Block diagram of the system to sense and record data from a typical wearable sensor.

**Figure 9 micromachines-15-00069-f009:**
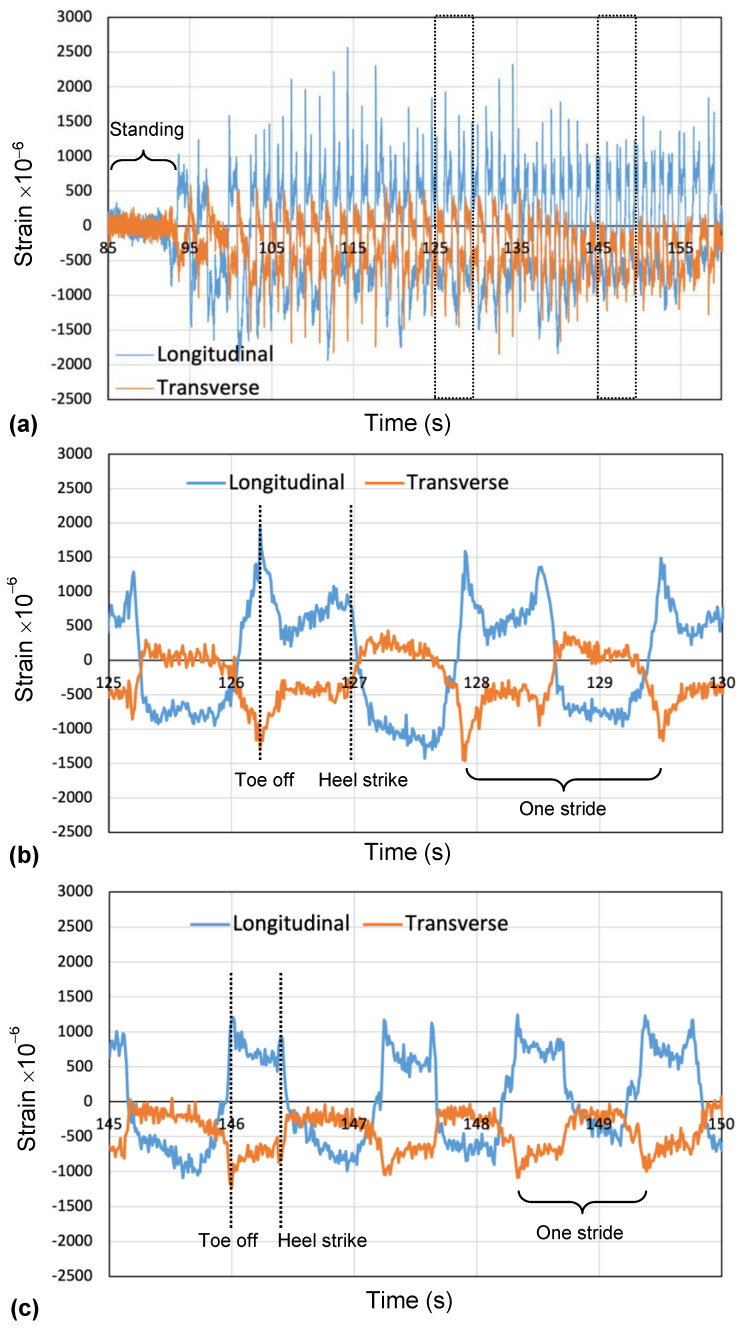
Typical example of deformation of toenail during walking on treadmill: (**a**) temporal changes during various movements, (**b**) regular walking, and (**c**) a small step taken during walking.

**Table 1 micromachines-15-00069-t001:** Cure and removal times of Materials A and B and cyanoacrylate on the different types of nails tested. The nails tested are a human nail (nail H), a white-colored acrylic nail (nail W), and a pink-colored acrylic nail (nail P).

**Cure times**			
**Adhesive**	**Nail H**	**Nail P**	**Nail W**
Material A—air cure	60 s	240 s	240 s
Material B—UV cure	60 s	60 s	60 s
Nail glue—air cure	20 s	none	20 s
**Removal times**			
**Adhesive**	**Nail H**	**Nail P**	**Nail W**
Material A—peel	<30 s	<60 s	<60 s
Material B—peel	<30 s	Could not remove	Could not remove
Nail glue—peel	Could not remove	NA	Could not remove

## Data Availability

All experiments were performed in accordance with institutional guidelines and regulations. The experimental protocol was approved by the IBM Research Institutional Review Board at the T.J. Watson Research Center, Yorktown Heights, N.Y., USA. All participants gave their informed consent to take part in the experiments. Two subjects participated in each of two experiments: one with sensors attached to their fingernails and the other with sensors attached to their toenails and walking on a treadmill. The data presented in this study are available on request from the corresponding author.

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
