# Peer review of "Healthcare Wearable Sensors Adhesion to Human Fingernails and Toenails"

_micromachines, 2023, doi:10.3390/mi15010069_

Round 1

Reviewer 1 Report

Comments and Suggestions for Authors

The study led by Katsuyuki Sakuma and team is titled "Biomedical Devices and Technologies in Intelligent Diagnosis: Theoretical Principles to Basic Applications." This research seamlessly aligns with the Micromechanics scope. I highly recommend its submission for publication in the Micromechanics Journal.

Author Response

We thank the reviewer for taking the time to critique our work.

Reviewer 2 Report

Comments and Suggestions for Authors

The article presents a pioneering approach to the adhesion of healthcare sensors to fingernails, introducing a novel method that holds promising implications for continuous wellness monitoring and disease tracking in everyday life. Focused primarily on the fingertip, a key point of interaction with the external world, the study explores the structure and techniques for attaching and removing these sensors, emphasizing the importance of meeting functional requirements, ensuring biological and environmental compatibility, and facilitating easy application and removal for both short- and long-term user applications. There are some issues that need to be addressed before publication:

1.     Compared with other related works, the innovation points of this paper need to be pointed out.

2.     The resolution of Fig.7 should be improved.

3.     In line 205, what is the physical or chemical mechanism of that protein may provide more energy for water molecules to escape the nail surface?

4.     In line 222, how do the autors define the difference between the mechanical bond and the chemical bond?

5.     In Fig.9, the y axis title 10-6 should be modified to 10e-6 or 10^-6

6.   In Fig.9, why the variation signal in longitudinal axis is much larger than the transverse axis and longitudinal axis

Comments on the Quality of English Language

Minor editing of English writing is needed.

Author Response

We thank the reviewer for taking the time to critique our work. Please see the attachment.

Reviewer 3 Report

Comments and Suggestions for Authors

The paper describes a novel approach for attaching a strain gauge sensor onto a human nail, employing a release layer material in conjunction with nail glue to enhance both sensor attachment and removal. These sensors, embedded in nails, hold promising potential for detecting various health issues and find application in diverse fields.

The study follows a formal and convenient methodology. While the investigation is somewhat constrained by the use of a relatively limited number of artificial nail materials for testing, coupled with minimal experimentation on human subjects, its novelty and potential results makes it interesting and significant from a research viewpoint. The manuscript, on the whole, is well-organized, although a section numbering error (6/7) is noted.

Nevertheless, Section 2, encompassing related works, should be expanded. In this section, the method of securing nail sensors in the located projects and works should be detailed and analysed.

Author Response

(The authors gave the same response as above.)

Round 2

Reviewer 2 Report

Comments and Suggestions for Authors

The authors have addressed the corresponding issues and I recommend its publication in its present form.